# Natural variation of *YELLOW SEEDLING1* affects photosynthetic acclimation of *Arabidopsis thaliana*

Roxanne van Rooijen [1,2,4], Willem Kruijer [3], René Boesten [1], Fred A. van Eeuwijk [3], Jeremy Harbinson [2] & Mark G.M. Aarts [1]

Exploiting genetic variation for more efficient photosynthesis is an underexplored route towards new crop varieties. This study demonstrates the genetic dissection of higher plant photosynthesis efficiency down to the genomic DNA level, by confirming that allelic sequence variation at the *Arabidopsis thaliana YELLOW SEEDLING1* (*YS1*) gene explains natural diversity in photosynthesis acclimation to high irradiance. We use a genome-wide association study to identify quantitative trait loci (QTLs) involved in the Arabidopsis photosynthetic acclimation response. Candidate genes underlying the QTLs are prioritized according to functional clues regarding gene ontology, expression and function. Reverse genetics and quantitative complementation confirm the candidacy of *YS1*, which encodes a pentatrico-peptide-repeat (PPR) protein involved in RNA editing of plastid-encoded genes (anterograde signalling). Gene expression analysis and allele sequence comparisons reveal polymorphisms in a light-responsive element in the *YS1* promoter that affect its expression, and that of its downstream targets, resulting in the variation in photosynthetic acclimation.

[1] Laboratory of Genetics, Wageningen University and Research, Droevendaalsesteeg 1, 6708 PB Wageningen, The Netherlands. [2] Horticulture and Product Physiology Group, Wageningen University and Research, Droevendaalsesteeg 1, 6708 PB Wageningen, The Netherlands. [3] Mathematical and Statistical Methods Group - Biometris, Wageningen University and Research, Droevendaalsesteeg 1, 6708 PB Wageningen, The Netherlands. [4] Present address: Cluster of Excellence on Plant Science, Heinrich Heine University, Düsseldorf, Germany. Correspondence and requests for materials should be addressed to M.G.M.A. (email: mark.aarts@wur.nl)

Light, as the driving force for photosynthesis, is a conspicuously important determinant of photosynthetic activity. Genetic variation exists in plants for what constitutes a high-light leaf or a low-light leaf, for the range of irradiances to which the leaf is capable of responding, and for the actual photosynthetic properties that emerge from any environmental treatment[1–5]. An increase in irradiance brings with it an increase in light-induced stress within the photosynthetic apparatus, with Photosystem II (PSII) being particularly sensitive to damage. To safely dissipate the extra absorbed light energy resulting from an irradiance increase, there is an immediate physiological response, comprising increases in photosynthetic metabolic activity, control of thylakoid electron transport, and the $q_E$ component of non-photochemical quenching[6]. If the increased irradiance level is sustained it will induce a slower acclimatory response via changes in the composition of mesophyll cells in terms of their proteins, pigments, and lipids, and other cofactors involved in electron transport and reactive-oxygen species metabolism[7–9]. The regulation of photosynthetic acclimation starts with signals originating from photoreceptors and from the photosynthetic machinery itself, going to the nucleus and altering patterns of nuclear gene expression (retrograde signalling)[9]. By identifying the genomic regions that associate with phenotypic variation before or after the increase in irradiance, or both, we can distinguish the regions that are associated with photosynthetic light use efficiency in general from those that are specifically associated with photosynthetic acclimation to an increase in irradiance[10,11]. Identifying the genes that give rise to the photosynthetic acclimation response will reveal at which regulatory level natural genetic variation for photosynthesis will act. Photosynthesis is a complex trait at both the physiological and genetic levels and as a result natural genetic variation in photosynthesis is an underused resource for identification of the genetic regulation of photosynthesis[5,12]. Together with the poor understanding of the complex relationship between photosynthesis and yield, this has resulted in photosynthesis being underused in plant breeding programmes[4,12]. Nevertheless it does have great potential for crop improvement[13–15].

In this study we investigate the natural genetic variation of *Arabidopsis thaliana* (Arabidopsis) for photosynthetic acclimation to an increased irradiance. Using genome-wide association analysis we first identify genetic quantitative trait loci (QTLs) associated with phenotypic variation among the Arabidopsis accessions we examined. We then explore the candidate genes underlying these loci to confirm which of them are causal for part of the observed variation. Our overall conclusion is that natural genetic variation of the *YELLOW SEEDLING1* gene explains one of the QTLs that affects photosynthetic acclimation to an increase in irradiance in Arabidopsis.

## Results

**Natural variation for photosynthetic acclimation**. Using chlorophyll fluorescence imaging, the light-use efficiency of PSII electron transport ($\Phi_{PSII}$) was measured in 344 Arabidopsis accessions at three time-points during the day, before and after the plants were subjected to a sudden increase in growth irradiance (Fig. 1a). The phenotypic distribution for $\Phi_{PSII}$ was narrow under steady low growth irradiance (100 µmol m$^{-2}$ s$^{-1}$), got broader upon high irradiance exposure (550 µmol m$^{-2}$ s$^{-1}$), and narrowed again during photosynthetic acclimation (Fig. 1b). These changes in the distribution of $\Phi_{PSII}$ were paralleled by the pattern of acclimation within individual plants, with young leaves acclimating faster than older (Fig. 1a). Following this increase in irradiance there was no visible leaf chlorosis or anthocyanin formation, although the index of chlorophyll content showed an increasing trend with time that was independent of irradiance and though there was an initial drop of anthocyanin content on the first day of high irradiance this was followed by a recovery throughout the acclimation period (Supplementary Fig. 1). Note that as anthocyanins absorb green wavelengths they will not significantly absorb the actinic wavelengths (660 nm) used in the measurement of $\Phi_{PSII}$[16]. All $\Phi_{PSII}$ measurements were highly positively correlated, indicating a coordinate regulation of photosynthesis in these irradiance environments (Supplementary Table 1). Broad-sense heritability varied between 0.06 and 0.09 prior to stress and between 0.20 and 0.33 after stress, marker-based estimates of narrow-sense heritability were close to zero prior to stress and varied between 0.30 and 0.52 after stress (Supplementary Table 2).

**Genetic variation associates with phenotypic variation**. Genome-wide association study (GWAS) analysis of $\Phi_{PSII}$ at each time point resulted in a pattern of associations, which is represented in a heat map of $-\log_{10}(p)$ values throughout the acclimation response (Fig. 2). We confined our attention to those QTLs identified by a SNP with a $-\log_{10}(p)$ association score of at least 4 and which recurred for at least three time points[5]. Thirty-four such QTLs were identified, and these were further classified according to the time and duration of their appearance. Eight QTLs were specific for the low-irradiance phase (low light, LL), 24 loci appeared after the onset of the high-irradiance treatment (high light, HL), of which six disappeared after a few days (early HL QTLs), and three occurred only after a few days (late HL QTLs). Two QTLs were present throughout the experiment, independent of the irradiance level (LL+HL; QTLs 8 and 9). We determined the physical positions of the SNPs corresponding to the QTLs associated with photosynthetic HL acclimation, and catalogued all genes in linkage disequilibrium (LD) with these SNPs. We measured LD in terms of $r^2$ corrected for genetic relatedness for each of the SNPs underlying a QTL (Supplementary Fig. 2)[17]. Because of differences in LD decay between QTLs, we decided to use fixed LD windows of 100 kb on either side of the top SNPs for candidate gene selection. This resulted in a list of 1531 candidate genes for which there may be allelic variation contributing to associated phenotypic variation (Supplementary Data 1). Only 51 of these scored positive for three out of five in silico selection criteria based on gene function that we used to prioritize the candidate genes: gene ontology, gene co-expression, gene expression in the vegetative rosette, and the presence of segregating polymorphisms in the coding sequence (Supplementary Data 2). These 51 priority candidate genes were associated with 19 of the acclimation-specific QTLs.

Given the structure of our phenotypic data as repeated measurements over time, alternative, multi-trait ways of performing a GWAS analysis offer themselves (Supplementary Data 3–5). First, given the single time point GWAS analyses, a smaller LD window around detected QTL was used, which resulted in 358 candidate genes (instead of 1531) amongst which were 10 of the 51 priority genes; second, we used a multi-trait analysis on principal components, which resulted in 2688 candidate genes amongst which were 23 of the 51 priority genes; and finally we used a multi-trait analysis across time points for $\Phi_{PSII}$ after the irradiance increase, which resulted in 1568 candidate genes, including 10 of the 51 priority candidates. The additional GWAS yielded many more candidate loci, but also confirmed many of the initially prioritized candidates. We therefore decided to retain this initial list of 51 candidate genes (Supplementary Data 2), and analyse T-DNA insertion knock-out mutant lines for these candidates.

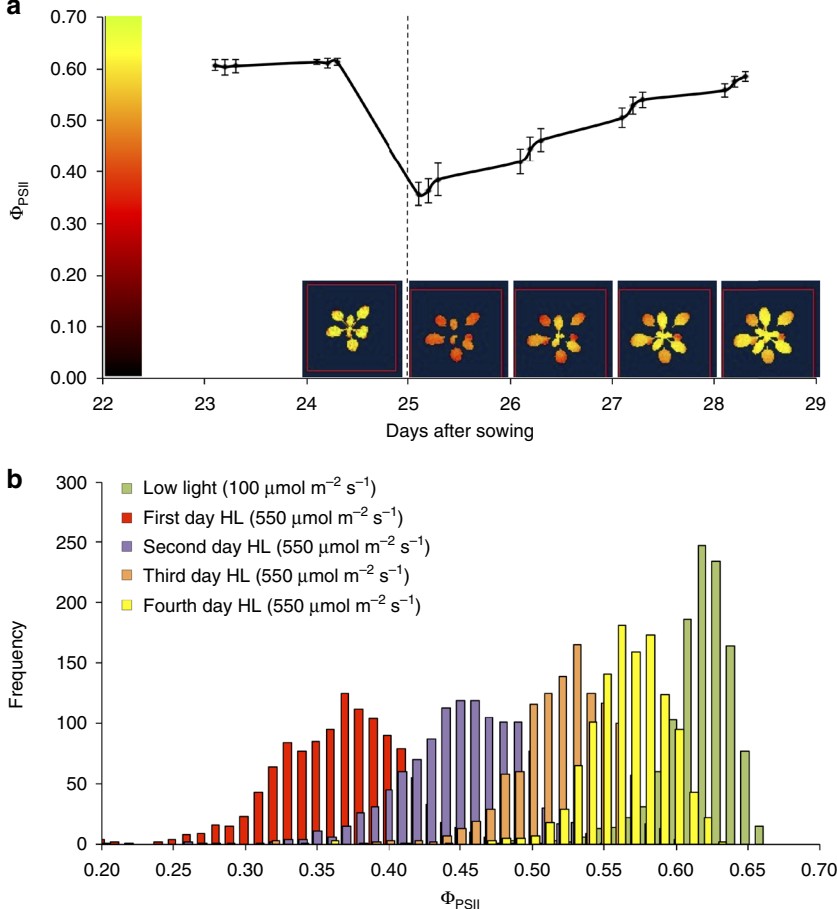

**Fig. 1** Photosynthesis efficiency of photosystem II in response to an increase in growth irradiance. **a** Photosynthetic acclimation of photosystem II ($\Phi_{PSII}$) of Arabidopsis accession Col-0. Shown are the $\Phi_{PSII}$ values over time ($\pm$standard error of the mean (s.e.m.); $N = 3$) and a chlorophyll fluorescence image of the same Col-0 plant measured 1 h after the beginning of the photoperiod, on five consecutive days. The vertical dotted line at 25 days after sowing indicates the increase in irradiance from 100 to 550 $\mu$mol m$^{-2}$ s$^{-1}$. **b** Frequency distribution of $\Phi_{PSII}$ measured for 344 Arabidopsis accessions at consecutive time points in the acclimation response to an increase in irradiance as shown in **a**, three replicate plants are measured per accession

**Exploring candidate genes**. T-DNA lines were tested for six genes: AT1G74180 (QTL16), AT3G04870 (QTL19), AT3G04880 (QTL19), AT3G22690 (QTL23), AT5G12290 (QTL31), and AT5G65010 (QTL34). These genes were chosen for two reasons. Either, because they belong to the strongest QTLs specific for HL, and of all genes in the LD regions of these QTLs the selected genes showed most promising polymorphisms segregating between extreme phenotypes based on haplotype analysis (Supplementary Fig. 3), or because they mapped to less strong HL-specific QTLs, but were annotated as being involved in photosynthesis efficiency and haplotype analysis of the gene showed promising polymorphisms segregating between extreme phenotypes (AT3G22690). When there was no priority gene with promising segregating polymorphisms within a strong HL-specific QTL, we chose the gene in which the significant SNP directly was located (AT5G12290). We recognize that in doing this we left many genes out of the current analysis, and these remain to be tested in future analyses.

For three selected genes (*YELLOW SEEDLING 1* (*YS1*; At3g22690), *DGD1 SUPPRESSOR 1* (*DGS1*; At5g12290), and *ASPARAGINE SYNTHETASE 2* (*ASN2*; At5g65010)) analysis of T-DNA insertion lines displayed an obvious aberrant phenotype for photosynthetic acclimation to increased irradiance compared to wild-type controls (Fig. 3). *YS1* was identified upon our initial GWAS approach; it encodes a pentatrico-peptide-repeat (PPR) protein involved in RNA editing of plastid-encoded genes[18] and

corresponds to QTL23 (Fig. 2). *DGS1* was detected in the first three GWAS approaches we tried; it encodes a mitochondrial outer membrane protein involved in galactolipid biosynthesis and corresponds to QTL31. *ASN2* was identified in all four GWAS approaches. It encodes an asparagine synthetase and corresponds to QTL-34.

**Variation at *YELLOW SEEDLING 1* affects $\Phi_{PSII}$ response**. Quantitative complementation did not confirm *DGS1* and *ASN2* as causal for the corresponding QTLs (Supplementary Figs. 4 and 5). This can easily happen if the phenotype is affected by epistatic interactions beyond these loci[19] so it does not exclude an important role for these genes in determining the light-response phenotype. We nonetheless subsequently focussed on *YS1*. Within the available re-sequence data for the Arabidopsis accessions used for the GWAS (http://1001genomes.org/), five different *YS1* alleles are distinguished (Fig. 4a). Accessions carrying alleles 2, 4, and 5 displayed the highest average photosynthesis efficiency in response to high light, and were significantly different from accessions with alleles 1 and 3 (Fig. 4b). Quantitative complementation of *YS1* was performed based on accessions carrying alleles 3 and 4, which indeed confirmed that the variation residing at QTL23 was due to allelic variation in the *YS1* gene (Fig. 4c). The quantitative complementation test was repeated with different accessions, carrying

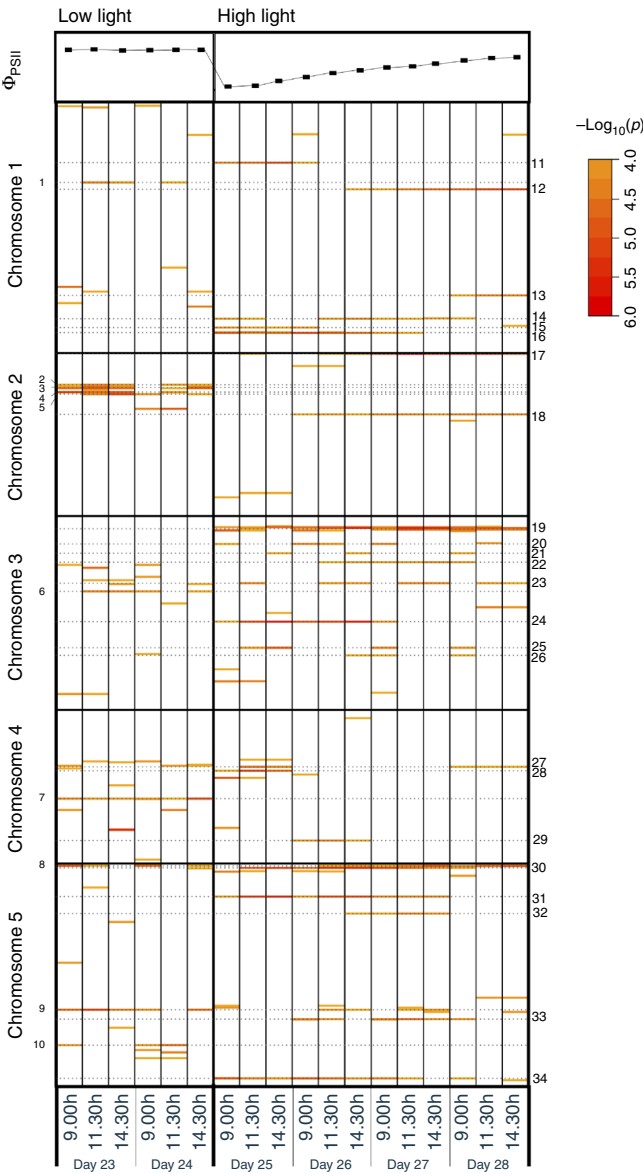

**Fig. 2** Heat map of SNP–ΦPSII association strengths as identified upon GWAS. A genome-wide association study (GWAS) of $\Phi_{PSII}$ is performed, before and after an increase in growth irradiance, based on a panel of 344 Arabidopsis accessions. Plant $\Phi_{PSII}$ (shown for Col-0 in the left panel) is measured three times per day, two days in low light (100 µmol m$^{-2}$ s$^{-1}$) and four days in high light (550 µmol m$^{-2}$ s$^{-1}$), as indicated in the right panel. The first measurement each day is taken 1 h after the beginning of the photoperiod. SNP–ΦPSII association strengths are reported as −log$_{10}$(p) values. The five Arabidopsis chromosomes are presented according to their physical size. Quantitative trait loci are numbered, 1–34. 1–10 are low-light specific, 11–34 are high-light specific

similar alleles, confirming the initial results (Supplementary Fig. 6). Analysis of the promoter DNA sequences of the Col-0 YS1-1 allele and the YS1-3 and YS1-4 promoter sequences (from five accessions each) identified three SNPs that distinguished YS1-1 from YS1-3 and YS1-4 (SNP[8024723], SNP[8025056], and SNP[8025189]). Of these only two were to be found in the public re-sequence data (Fig. 4a and Supplementary Fig. 7). In addition, between positions 8,024,863 and 8,024,871 bp there was an 8-bp deletion in the YS1-3 promoter (InDel[8024863–8024871]) which was also not to be found in the public re-sequence data (Supplementary Fig. 7). This InDel overlaps with a binding site for the nuclear transcription factor GT-1, while SNP[8025189] is located in the core sequence of another GT-1 binding site (Supplementary Fig. 7)[20,21]. Mutations in GT-1 binding sites are known to affect the responsiveness of promoters to light[22,23]. No transcription factor binding sites were found around SNP[8024723] or SNP[8025056].

## Discussion

A *ys1-1* knock-out mutation of *YS1-1* has been found to lead to disturbed chloroplast development in young seedlings[18], and impaired repression of LHCB genes in response to increased growth irradiance[24]. These are thought to be due to differences in editing of chloroplast-encoded RNA transcripts[18]. Mutation of *YS1-1* also affects the editing of *rpoB* transcripts. *RpoB* encodes the β-subunit of the Plastid-Encoded Polymerase (PEP) protein[18,25]. PEP is mainly involved in transcription of chloroplast-encoded photosynthesis genes encoding photosystem I (PSI) and PSII components and chloroplast-encoded tRNAs[26]. PEP-transcribed chloroplast-encoded tRNAs are important for normal photosynthesis[27]. We now show that *YS1* is also involved in the response to an increase in irradiance. This involvement is likely to be due to differences in expression of chloroplast genes mediated through altered PEP activity. To confirm this, we measured the

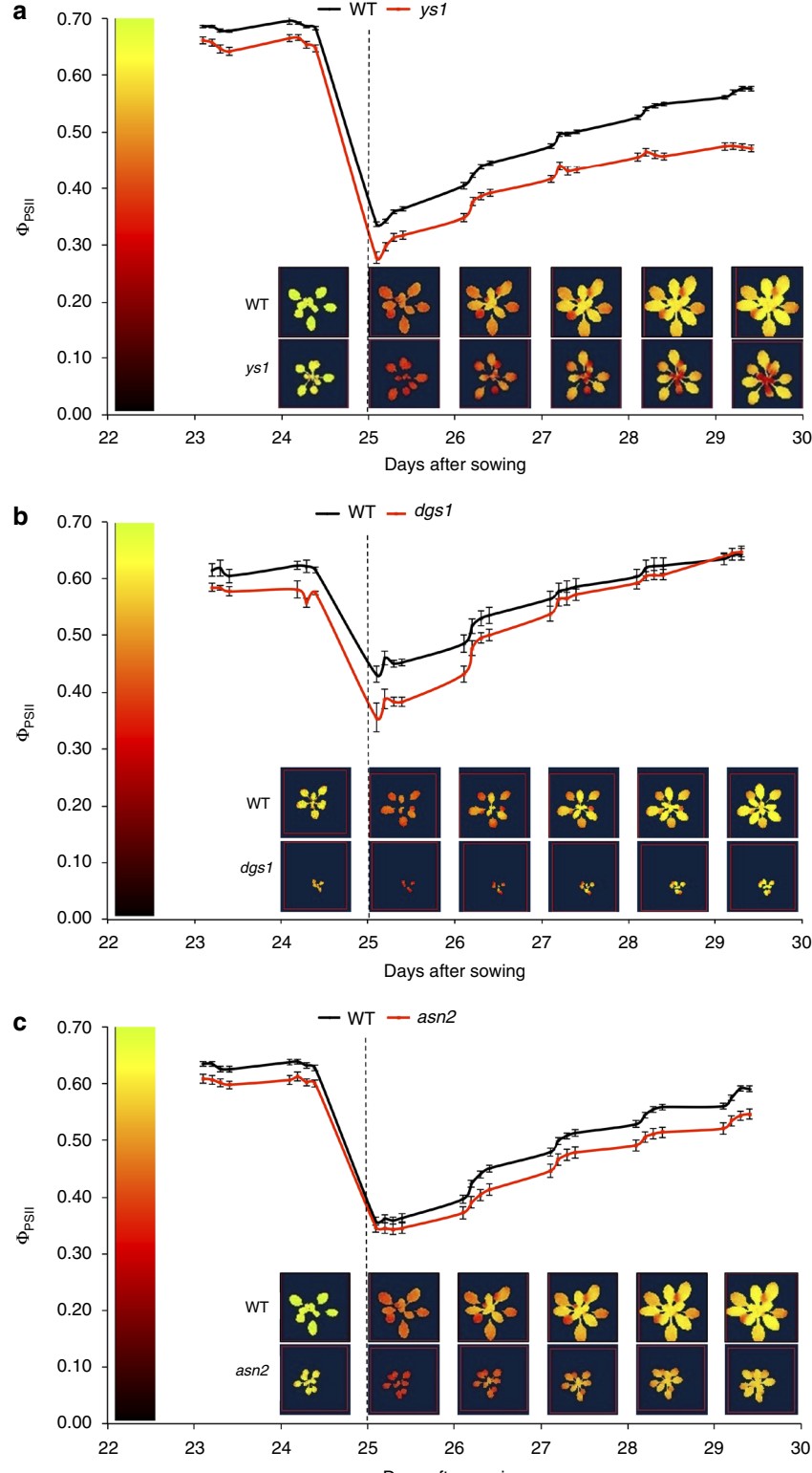

**Fig. 3** Candidate gene mutant photosynthesis efficiency response to increased irradiance. **a–c** Photosynthesis efficiency ($\Phi_{PSII}$) response over time (±s.e.m., $N = 16$) and chlorophyll fluorescence images of knock-out mutants for the **a** *YELLOW SEEDLING 1* (*ys1*); **b** *DGD1 SUPPRESSOR 1* (*dgs1*); and **c** *ASPARAGINE SYNTHETASE 2* (*asn2*) gene, compared to Col-0 wild-type plants (WT). The chlorophyll fluorescence images were taken 1 h after the beginning of the photoperiod. The vertical dotted line at 25 days after sowing indicates the increase in irradiance from 100 to 550 µmol m$^{-2}$ s$^{-1}$

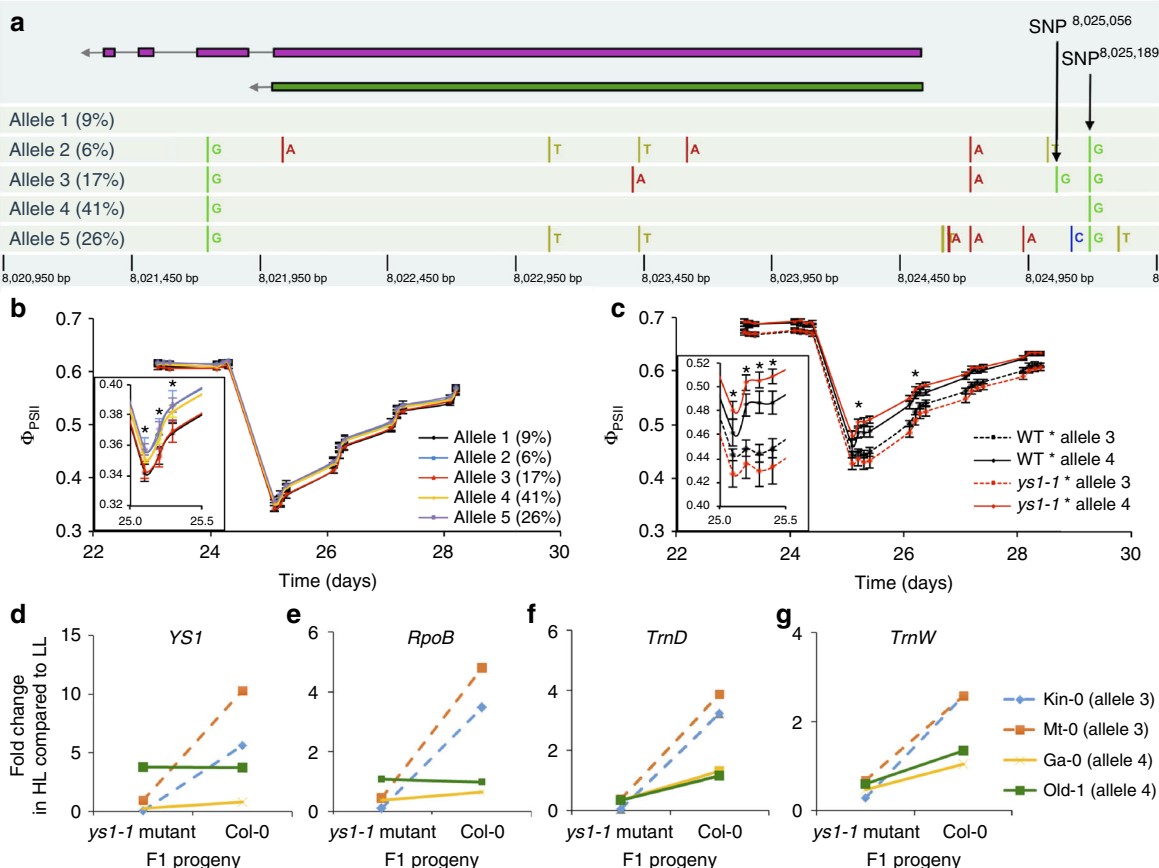

**Fig. 4** Characterization of natural alleles of *YS1*. **a** Schematic representation of the five most abundant *YS1* allelic haplotypes present in the GWAS panel and their frequencies (%) (www.arabidopsis.org). NB gene orientation is 3′ to 5′ as indicated with an arrow on the left; two splice variants are known (purple and green), exons are indicated with purple or green boxes and introns with lines connecting them. SNPs differing from the Col-0 reference genome sequence (*YS1* allele 1; *YS1-1*) are marked; two SNPs referred to in the text are indicated by arrows, chromosome 3 positions are indicated at the bottom (picture from http://signal.salk.edu/atg1001/3.0/gebrowser.php). **b** Average photosynthesis efficiencies ($\Phi_{PSII}$) ($\pm$s.e.m.) of accessions carrying one of five *YS1* alleles before (LL) and after an increase in irradiance (HL), the inset shows small but significant differences (*$p < 0.05$, *T*-test series) between allelic groups. **c** $\Phi_{PSII}$ ($\pm$s.e.m.; $N = 16$) of F1 progeny of crosses between Col-0 wild type (WT; *YS1-1*) or *ys1-1* mutants with either Ll-0 carrying a *YS1-3* allele, or Ga-0 carrying a *YS1-4* allele, confirming that natural genetic variation at the *YS1* gene explains the phenotypic variation identified by QTL23; * indicates a significant difference ($p < 0.05$; *T*-test) in allelic effect on the $\Phi_{PSII}$ difference at indicated time points between the different WT × accession and *ys1* × accession F1 progenies. **d**–**g** Fold changes in mRNA expression of *YS1* (**d**), *RpoB* encoding the β-subunit of the Plastid-Encoded Polymerase (PEP) protein (**e**), and the chloroplast-encoded tRNA genes *TrnD* (**f**) and *TrnW* (**g**), two transcriptional targets of PEP, transcription of which is affected by *YS1*. Expression at HL is compared with expression at LL in F1 progeny of crosses between either Col-0 WT or the Col-0 *ys1-1* mutant and natural accessions Kin-0 and Mt-0 (carrying the *YS1-3* allele, dashed lines) or Ga-0 and Old-1 (carrying the *YS1-4* allele, continuous lines)

expression of the *YS1* gene, the *rpoB* gene and the chloroplast tRNA genes *trnD* and *trnW* in F1 plants produced by crossing Col-0 *YS1-1* or *ys1-1* to accessions carrying *YS1-3* or *YS1-4* alleles. Differences in expression of *YS1*, *rpoB*, *trnD*, and *trnW* were found between accessions carrying *YS1-3* or *YS1-4* when they were heterozygous with the wild-type or with the mutant allele of *YS1-1* (Fig. 4d–g). Analogous to the strong effect conferred by the knock-out *ys1-1* allele, the different functional alleles of *YS1* affect the expression of these chloroplast genes, but in a more subtle way. The *YS1-4* allele is the most dominant of all, with there being little difference between *YS1-1*/*YS1-4* and *ys1-1*/*YS1-4* genotypes, while *YS1-1* is dominant over *YS1-3* based on the differences in expression between the *YS1-1*/*YS1-3* and *ys1-1*/*YS1-3* genotypes. This agrees well with the allelic differences in the *YS1* promoter, with the *YS1-3* allele missing one of the GT-1 binding sites (InDel[8024863–8024871]) and the *YS1-4* allele carrying an alteration in another GT-1 binding site which, when compared to the *YS1-1* allele (SNP[8025189]), may improve its affinity for GT-1. In addition to the effect of the *YS1-3* and *YS1-4* alleles on the expression of chloroplast-encoded genes, it is interesting to study their effect on

photosynthetic acclimation of leaves of different developmental stages, specifically when knowing the function of YS1 in chloroplast development. In the *ys1-1* knock-out mutant, young, newly emerging leaves are poorly acclimating throughout the experiment (Fig. 3). When comparing natural accessions, especially old leaves acclimate faster in plants carrying the *YS1-4* allele than in plants carrying the *YS1-3* or *YS1-1* alleles, whereas acclimation in the young, newly emerging leaves is not different (Supplementary Fig. 6). The allelic differences can be best examined in the F1s obtained from crossing the *ys1-1* mutant to accessions carrying either *YS1-4* or *YS1-3*. These F1s confirm the faster acclimation of old leaves when the *YS1-4* allele is present, and also show this effect for young leaves (Supplementary Fig. 6).

It again illustrates the superior effect of the *YS1-4* allele on acclimation to high irradiance, compared to the other functional alleles, probably through improved affinity of its promoter for GT-1. While the allelic difference was not obvious for chlorophyll reflectance, it did show up for the anthocyanin reflectance index. Also for anthocyanin reflectance, the *YS1-4* allele is the strongest one, dominant over the *YS1-1* allele of Col-0, which itself is

dominant over the *YS1-3* allele (Supplementary Fig. 6f, g). Increased anthocyanin formation thus appears to be a pleiotropic effect of *YS1* function that very well may contribute to the ability to acclimate best to high irradiance.

The different effects of natural alleles of a gene affecting photosynthetic properties illustrate the potential for improved photosynthetic properties. While we demonstrate this for the model species *Arabidopsis thaliana*, there is no reason to assume this would not be feasible for crop species, and this thus opens the way to initiate breeding programs for improved photosynthesis traits as a way to improve crop yield.

## Methods

**Experimental design**. A set of 344 *Arabidopsis thaliana* (Arabidopsis) accessions was used for GWAS, which are all part of a core set of 360 natural accessions that represent the global genetic diversity of the species (https://www.arabidopsis.org/servlets/TairObject?type=stock&id=4501958598)[28]. Sixteen accessions of the core set were not used: CS28051, CS28108, CS28808, CS28631, CS76086, CS76104, CS76110, CS76112, CS76118, CS76196, CS76212, CS76254, CS76257, and CS76302. The T-DNA lines studied were: SALK_123515 (for YS1), SAIL_391_F04 (for DGS1), and SALK_043167 (for ASN2).

Plants were grown hydroponically in a climate controlled growth chamber, on rockwool blocks (Grodan Rockwool Group, $40 \times 40 \times 40$ mm in size) supplied with a nutrient solution developed for Arabidopsis (pH 7; EC 1.4 mS cm$^{-1}$) consisting of 1.7 mM NH$_4^+$, 4.5 mM K$^+$, 0.4 mM Na$^+$, 2.3 mM Ca$^{2+}$, 1.5 mM Mg$^{2+}$, 4.4 mM NO$_3^-$, 0.2 mM Cl$^-$, 3.5 mM SO$_4^{2-}$, 0.6 mM HCO$_3^-$, 1.12 mM PO$_4^{3-}$, 0.23 mM SiO$_3^{2-}$, 21 µM Fe$^{2+}$ (chelated using 3% diethylene triaminopentaacetic acid), 3.4 µM Mn$^{2+}$, 4.7 µM Zn$^{2+}$, 14 µM BO$_3^{3-}$, 6.9 µM Cu$^{2+}$, and <0.1 µM MoO$_4^{4-}$, and at a constant irradiance of 100 µmol m$^{-2}$ s$^{-1}$ (Philips 610 fluorescent tubes, MASTER TL5 HO, 80W)[5]. The photoperiod was set to 10 h/14 h day/night, temperature was set to 20/18 °C (day/night), relative humidity was set at 70% and CO$_2$ levels were ambient. On day 25 after sowing the irradiance was increased to 550 µmol m$^{-2}$ s$^{-1}$ at the onset of the photoperiod. Seeds were pre-sown on filter paper in petri dishes wetted with 0.5 ml demineralized water, and placed in the dark at 4 °C for 4 days to stratify. Once stratified, the seeds were planted on the rockwool blocks. The blocks were positioned and secured using a frame consisting of a baseplate made from a sheet of perforated stainless steel, a second PVC frame that was held 15 mm above the stainless steel base and into which the blocks were placed, and a black non-reflective foamed PVC cover-sheet drilled with countersunk holes 60 mm apart and 3 mm diameter that were positioned over the centres of the blocks. The baseplate was supported 5 mm above the floor of the basin, allowing nutrient solution to pass freely and uniformly under the growing frame and circulate through the frame via the holes in the perforated metal baseplate and the 10-mm spacers between the blocks.

**Chlorophyll a fluorescence imaging and analysis**. The responses of photosynthesis, whether short-term or long-term, to an irradiance increase can be conveniently monitored using chlorophyll fluorescence, from which parameters quantifying the light-use efficiency of PSII (i.e., linear) electron transport and other aspects of PSII operation and regulation can be derived. In particular in the absence of a stress that causes chlorosis due to a loss of leaf photosynthetic pigmentation, an increase in the light-use efficiency of PSII electron transport ($\Phi_{PSII}$) under conditions of unchanging irradiance would be consistent with an increase in assimilation of carbon dioxide. Chlorophyll a fluorescence has been used in the past to identify photosynthesis mutants[29]. In this work we use chlorophyll fluorescence to phenotype naturally occurring variation for photosynthetic traits as a first step in a genome-wide analysis. Chlorophyll a fluorescence was measured using a high-throughput phenotyping system developed for Arabidopsis[5,30]. Chlorophyll fluorescence was measured at 730 nm and excited using radiation from Phlatlight LEDs (Luminus, Billerica, Massachusetts, USA) (peak emission wavelength 624 nm). In all experiments, the photoperiod lasted from 8.00 h until 18.00 h CET and imaging of light use efficiency of photosystem II ($\Phi_{PSII}$) was performed daily at 9.00 h, 11.30 h, 14.30 h (and 16.30 h). Imaging was measured on three time points per day (9.00 h, 11.30 h, and 14.30 h); in all further experiments plants were measured on four time points per day (9.00 h, 11.30 h, 14.30 h, and 16.30 h). Indices of leaf chlorophyll (both chlorophyll a and b) and anthocyanin contents was estimated calculated using leaf reflectance measurements, with reflectance at 700 and 790 nm being used to estimate chlorophyll content and reflectance at 550 and 700 nm being used to estimate anthocyanin content[16,30,31]. These measurements were made once per day.

**Genome-wide association analysis**. Using a mixed model[32], GWAS analyses were performed for each time point, using the mean $\Phi_{PSII}$ for each accession and 214,051 SNPs[33], of which we used the 199,589 SNPs with a minor allele frequency of at least 0.05. The mixed model was tested to adequately account for potential confounding by drawing QQ-plots for each time measurement of $\Phi_{PSII}$ (Supplementary Fig. 8). SNPs were classified as QTLs whenever their -log($p$) value $\geq 4$ for

at least three of the low-irradiance time-points, or for at least three of the high-irradiance time-points. Genes within 100 kb of these SNPs were listed as candidate genes, and QTLs with overlapping windows were lumped together to form one QTL. QTLs were numbered according to physical position, and identified according to time of appearance, i.e., only in low light, early in the response to high light, late in the response to high light, or at all time points.

**Single and multi-trait GWAS analysis**. Four different GWAS analyses were performed for the high-light time-points. The first being the analysis as presented in the main text: single time-point GWAS, selecting SNPs associated with $\Phi_{PSII}$ ($p < 0.0001$, $T$-test) for at least three time-points, and genes in a 100-kb window around these SNPs. The second being the single time-point GWAS analysis, selecting SNPs associated with $\Phi_{PSII}$ ($p < 0.0001$, $T$-test) for at least three time-points, and genes in a 20-kb window around these SNPs (i.e., the same analysis as in the main text, but with smaller LD windows). The third being the multi-trait GWAS analysis on the first four principal components based on the standardized genotypic means of the 12 high-light time-points. Multi-trait GWAS was performed using the GEMMA software[34], which assumes a multi-trait mixed model (MTMM) containing both genetic and environmental correlations. For each SNP in turn we tested the null-hypothesis that there is no SNP-effect on any of the traits, the alternative being that at least one SNP effect is non-zero. To select candidate genes we used 100-kb windows around SNPs associated with $\Phi_{PSII}$ ($p < 0.0001$, $T$-test). The fourth being the multi-trait GWAS analysis[34] on curve parameters. We tested three curve parameters: the maximum increase in $\Phi_{PSII}$ over time, i.e., the largest slope ($\Phi_{PSII}$ $(t_2)-\Phi_{PSII}$ $(t_1))/(t_2-t_1)$, where the maximum is taken over all high-light time-points $t_1<t_2$ that are on different days; the early recovery: the largest $\Phi_{PSII}$ value observed during the second day in high-light conditions, minus the lowest $\Phi_{PSII}$ value observed during the first day in high-light conditions; and the late recovery: the largest $\Phi_{PSII}$ value observed during the third and fourth day in high-light conditions, minus the largest $\Phi_{PSII}$ value observed during the second day in high-light conditions. These parameters were first determined for each individual plant, and the GWAS was based on accession means of these parameters. Again we used 100-kb windows around SNPs associated with $\Phi_{PSII}$ ($p < 0.0001$, $T$-test) to select candidate genes. All analyses were restricted to SNPs with minor allele frequency larger than 0.05. GWAS analyses 1–4 gave 1531, 358, 2688, and 1568 candidate genes, respectively.

**Candidate gene prioritization**. All 1531 candidate genes were prioritized in an in silico analysis using publicly available databases for five criteria. The first criterion was based on gene ontology terms (www.arabidopsis.org); candidate genes with ontology terms "chloroplast", "photosynthesis", and "light stress" were identified. The second criterion was based on gene co-expression profiles[35] (www.genevestigator.com); genes whose expression correlated with the expression of light-response genes with an $r^2 > 0.8$ were identified. The light-response genes were taken from light intensity-associated microarray experiments. The third criterion was based on the presence of gene expression in the vegetative rosette[36] (http://bar.utoronto.ca/); candidates genes with any expression in the vegetative rosette were identified. The fourth criterion was based on annotated function; candidate genes known to have a function related to photosynthesis based on literature were identified. The fifth criterion was based on the presence of polymorphisms in candidate genes that segregated between two groups of 15 accessions with the most extreme phenotypes (http://1001genomes.org/). Whenever a candidate gene scored positive for three out of these five criteria, it was included in the priority candidate list (Supplementary Data 2).

**Haplotype analysis**. Haplotypes, representing natural alleles, were assigned based on all SNPs in the promoter and coding regions of candidate genes using the re-sequence data of 173 accessions (Supplementary Table 3). Those haplotypes that occurred in > 4% of the 173 accessions were then associated with photosynthetic phenotypes. Haplotypes that resulted in different photosynthetic response to increased irradiance (based on two-sided Student's $t$-test) were selected for quantitative complementation tests.

**Quantitative complementation**. Quantitative complementation was used to confirm the contribution of allelic difference at one locus to the observed phenotypic variation[37]. This was performed by crossing two accessions with different alleles for the gene involved, to a T-DNA insertion knock-out mutant for the gene, in accession Columbia-0 (Col-0) background, as well as to the Col-0 wild type (both used as maternal line). The phenotype of resulting F1 plants ($N = 16$ per cross) for their photosynthetic response to increased irradiance was determined as described above. Two-way ANOVA was performed to test if the difference in photosynthesis response to light increase between the F1 plants of Col-0 and *ys1-1* to an accession homozygous for the *YS1-3* allele, was significantly different ($p < 0.05$) from the difference in photosynthesis response to light increase between the F1 plants of Col-0 and *ys1-1* to an accessions homozygous for the *YS1-4* allele. For quantitative complementation of *YS1* we used accession CS76172, which is homozygous for the *YS1-3* allele, and accession CS76133, which is homozygous for the *YS1-4* allele. To confirm the results of this first complementation test, the experiment was checked by making new crosses of Col-0 and *ys1-1* to three

different accessions homozygous for the YS1-3 allele (CS76153, CS76192, and CS76218), and comparing them to new crossings of Col-0 and ys1-1 to two different accessions homozygous for the YS1-4 allele (CS28583 and CS76133).

**Quantitative reverse transcription PCR**. At time point 11.00 a.m. CET (i.e., 3 h after lights on) on days 24 (LL plants) and 25 (HL plants) after sowing, whole rosettes were collected and flash-frozen in liquid nitrogen. RNA was isolated according to Onate-Sánchez and Vicente-Carbajosa[38]. After normalization of RNA concentrations, cDNA was synthesized using the Iscript cDNA synthesis kit (Bio-RAD, www.bio-rad.com). qRT-PCR was performed with three technical replicates for each biological replicate using the SYBR-green master mix (Bio-RAD, www.bio-rad.com). Three biological replicates were used per accession, four accessions were analysed per haplotype. Two reference genes were used for normalization: *UBI-QUITIN7* (*UBQ7*; At2g35635) and *CYTOCHROME B5 ISOFORM E* (*CB5E*; At5g53560); transcription levels of *UBQ7* and *CB5E* were shown to be constant under excess light[39,40]. The primers used for qRT-PCR are listed in Supplementary Table 4. We chose the accessions CS76113, CS28193, CS28492, and CS76297 to represent allele 1; the accessions CS76305, CS28685, CS76218, and CS76153 to represent allele 3; and the accessions CS28787, CS76133, CS76129, and CS76128 to represent allele 4. One-way ANOVA was used for testing significance. The changes in expression were calculated for each genotype by averaging the gene expression at LL conditions (LL average) and then determining the fold difference for the HL expression compared to the LL average.

**Data availability**. All relevant data are available from the authors upon request.

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

## Acknowledgements

We thank Prof Dr Maarten Koornneef, Dr Pádraic Flood, and Dr Aina Prinzenberg for critical reading of the manuscript; Prof Dr Christoph Benning for his kind donation of the knock-out line for *DGS1*; and Dr Akira Suzuki for his kind donation of the knock-out line for *ASN2*. This project was carried out within the research programme of BioSolar Cells, co-financed by the Dutch Ministry of Economic Affairs. W.K. was partially funded by the Learning from Nature project of the Dutch Technology Foundation (STW), which is part of the Netherlands Organisation for Scientific Research (NWO).

## Author contributions

R.v.R. and R.B. propagated, genotyped, and screened material. R.v.R., J.H. and M.G.M.A. designed the study. R.v.R., W.K., R.B., F.A.v.E., J.H., and M.G.M.A. analysed data, discussed the results and wrote the paper.

## Additional information

**Competing interests:** The authors declare no competing financial interests.

