## [Peer Review File · Nature Communications]

Editorial Note: This manuscript has been previously reviewed at another journal that is not operating a transparent peer review scheme. This document only contains reviewer comments and rebuttal letters for versions considered at Nature Communications. Mentions of prior referee reports have been redacted.

PEER REVIEW FILE

Reviewers' Comments:

Reviewer #1 (Remarks to the Author):

The authors have satisfactorily revised the manuscript and performed additional experiments which confirm that allelic variations in the YSI promoter lead to different levels of RNAs of chloroplast genes transcribed by the PEP polymerase. This has significantly improved the manuscript.

Reviewer #2 (Remarks to the Author):

In this submission, the authors use an interesting approach to addressing an important topic in plant biology, which has important implications for future crop breeding. I am not qualified to comment on the genetic aspects so my comments are restricted to the phenotyping and physiology.

The gene identified using this approach, YS1, is not novel, having been identified as having a role in chloroplast development. The phenotype shown for the mutant of this gene is consistent with this in that maturer leaves appear to acclimate, whilst newly emerging leaves are poorly acclimated. This may reflect a fault in early chloroplast development that becomes apparent at the higher irradiance only. No analysis of this is carried out however. It would be interesting to see an analysis distinguishing leaves at different developmental stages, which maybe possible from the existing dataset.

In responding to my previous review of this paper, I think the authors did not always understand what I was getting at. In particular, I raised the point before that an increase in PSII efficiency may reflect either an increase in capacity for photosynthesis (i.e. increased carbon assimilation) or decreased light harvesting (reduced antenna size) and that they have not distinguished these. In the revised m/s they do state that chlorophyll content and anthocyanins did not change in

response to increased light (lines 62-3) but no data seem to be present to support this assertion. It is not possible to judge by eye whether there has been a 10% change in chlorophyll content (which would explain the response seen. They authors are not able to say from the data presented whether the response they see in chlorophyll fluorescence related to an increase in photosynthetic capacity or a reduction in antenna size. Some relatively simple in vivo measurements (gas exchange, fluorescence induction, pigment analysis) should be sufficient to clarify this and add to the value of the physiological work.

Reviewer #5 (Remarks to the Author):

The manuscript reports the identification of candidate genes underlying the natural variation of photosynthesis efficiency in a diverse set of 344 *Arabidopsis thaliana* accessions. Photosynthetic efficiency was characterized through chlorophyll fluorescence imaging before and after a sudden increase in light irradiance. A pretty large number of candidate genes were reported by an initial GWAS analysis and the list was first narrowed down to 51 genes and then to 6 genes by making use of several criteria such as their expression and co-expression or their ontology and sequence variability between extreme phenotypes. The effect of 3 of these 6 genes was further confirmed using T-DNA lines which displayed a phenotype related to photosynthesis efficiency. In addition, one of these 3 genes, YS1, was validated by quantitative complementation. Finally, expression analyses and allele sequence comparisons further confirmed the effect of YS1 and pointed out sequence variation in its promoter that may account for the observed phenotypic variation.

The paper is interesting, it represents a large experimental effort and the results are pretty much convincing. I thus think they deserve to be published. Nevertheless I have some concerns with the way the manuscript is written because sometimes the story seems strange. Part of this may be due to some changes following the first round of review, especially with respect to the GWAS part.

Indeed, I do believe that the multivariate methods are more appropriate to dynamically study a phenotype (which appeared to be on originality of the present paper with respect to previous work from the same team) than the univariate ones, but unfortunately the results from the multivariate analyses are not really well integrated to the manuscript as underlined by the fact that the initial list of 51 candidate genes was not modified following the multivariate analyses. One of the reason for this is likely because the gene YS1 was found only with the univariate approach and since the paper is focusing on this gene it may be difficult to account for the multivariate analyses results. Anyhow, I think this is a problem because it does not make the things easy to follow. Maybe this issue should be acknowledged and discussed in the manuscript.

Also, YS1 was not found with a smaller window than 100 kb around the associated SNPs (method 2) because it is quite far away (100 kb) from the significant SNPs in QTL 23. I think

that 100 kb on each side of the top SNP, which actually yields 200 kb QTL windows, is quite large given the expected LD decay of 10 kb in *Arabidopsis thaliana* (Kim et al., 2007). I admit that LD varies quite a lot in the genome especially because of natural selection but no information is given in the manuscript about LD extent around the QTLs and I clearly doubt that the LD decay would be the same within each of the 34 QTLs detected. Maybe LD decay could be estimated within each QTL to support the choice of 200 kb. If the authors want to underline their global approach as an original way to identify genes that matter for natural phenotypic variation, they should at least discuss these points.

Finally, it is also unfortunate that the QTL 23 which harbor YS1 do not really show a dynamic response since it is detected both in low and high light (figure 2). If this is somehow consistent with the T-DNA results (figure 3), this is not really illustrative of the dynamic nature of the phenotype. Again, I do not have any fundamental problem with this but I think this should be mentioned and discussed in the text and maybe the goal of the paper should be modified.

Minor points:

- In the abstract it is stated “we show that it is feasible to dissect natural variation in photosynthesis efficiency down to the genomic DNA level by confirming that allelic sequence variation at the YELLOW SEEDLING1 (YS1) gene explains natural variation...”. But in the end, how much of the natural variation is explained by YS1 alleles?
- I would be interested by more quantitative genetics on the phenotypes at each time point. What are their broad and narrow (marker-based) heritability? Is this phenotype structured within the population? Is there any correlation with the latitude of origin of the accessions? If so, are the alleles structured within the population under study? If this is the case I would expect some LD within the causal region which would support the choice of 200 kb for picking the candidate genes.
- Related to the potential structure of the phenotypes under study, I would be interested to see a QC of the GWAS through for instance QQ-plots or p-value distribution just to make sure that the potential confounding was well accounted for by the mixed-model.
- There seems to be three time-replicates for GWAS and four time-replicates for F1, but in the material and methods this distinction is not made?
- Epistasis is postulated as a reason for the non validation of DGS1 and ASN2 by quantitative complementation. Do you have some data/results to support this statement?

Revision of manuscript NCOMMS-16-29548A "A regulator in anterograde signalling underlies natural variation for plant photosynthesis" by van Rooijen et al.

Response to Reviewers' comments:

Reviewer #1 (Remarks to the Author):

The authors have satisfactorily revised the manuscript and performed additional experiments which confirm that allelic variations in the YS1 promoter lead to different levels of RNAs of chloroplast genes transcribed by the PEP polymerase. This has significantly improved the manuscript.

⇒ *Response: We were very pleased to read this, thanks.*

Reviewer #2 (Remarks to the Author):

In this submission, the authors use an interesting approach to addressing an important topic in plant biology, which has important implications for future crop breeding. I am not qualified to comment on the genetic aspects so my comments are restricted to the phenotyping and physiology.

The gene identified using this approach, YS1, is not novel, having been identified as having a role in chloroplast development. The phenotype shown for the mutant of this gene is consistent with this in that maturer leaves appear to acclimate, whilst newly emerging leaves are poorly acclimated. This may reflect a fault in early chloroplast development that becomes apparent at the higher irradiance only. No analysis of this is carried out however. It would be interesting to see an analysis distinguishing leaves at different developmental stages, which maybe possible from the existing dataset.

⇒ *Response: We appreciate this suggestion and did extra analysis of chlorophyll fluorescence false colour images of individual leaves of different developmental stages and added the results in the Supplementary Figure S4. It gave interesting extra results regarding the effect of the different natural alleles of YS1 that confirm the results that we already showed. When comparing natural accessions, especially old leaves acclimate faster in plants carrying the YS1-4 allele than in plants carrying the YS1-3 or YS1-1 alleles, whereas acclimation in the young, newly emerging leaves is not different (Fig. S4). The F1s examined in the quantitative complementation experimentation confirm the faster acclimation of old leaves when the YS1-4 allele is present, and also show this effect for young leaves (Fig. S4).*

In responding to my previous review of this paper, I think the authors did not always understand what I was getting at. In particular, I raised the point before that an increase in PSII efficiency may reflect either an increase in capacity for photosynthesis (i.e. increased carbon assimilation) or decreased light harvesting (reduced antenna size) and that they have not distinguished these. In the revised m/s they do state that chlorophyll content and anthocyanins did not change in response to increased light (lines 62-3) but no data seem to be present to support this assertion. It is not possible to judge by eye whether there has been a 10% change in chlorophyll content (which would explain the response seen). They authors are not able to say from the data presented whether the response they see in chlorophyll fluorescence related to an increase in photosynthetic capacity or a reduction in antenna size. Some relatively simple in vivo measurements (gas exchange, fluorescence induction, pigment analysis) should be sufficient to clarify this and add to the value of the physiological work.

⇒ *Response: We have added data on chlorophyll and anthocyanin content based on leaf reflectance data (Supplementary Figure S4). This analysis again gave interesting extra results regarding the effect of the different natural alleles of YS1 that confirm the results that we already showed. No allelic difference was obvious for chlorophyll reflectance index, whereas for anthocyanin reflectance, the YS1-4 allele again shows to be the strongest one, dominant over the YS1-1 allele of Col-0, which itself is dominant over the YS1-3 allele (Fig S4 - F and G). Note, however, that the 660 nm actinic irradiance used to measure Φ_{PSII} will not be absorbed by anthocyanins. The leaf reflectance data used to estimate chlorophyll content, are, we believe, more important in dealing with uncertainties raised by the reviewer. The reflectance method used (Gitelson et al.; 2003) compares a 'red-edge' reflectance at 700 nm with a near-infra red reference wavelength at 790 nm. The lack of any allelic differences in reflectance parameter implies that leaf-light absorption in the red region of spectrum (which is due to chlorophylls) did not change*

differentially during the treatment (there was an weak overall upwards trend in the chlorophyll index). If light absorption does not change then the possible cofounding effect of changes in light absorption on the interpretation of the Φ_{PSII} data is removed.

Reviewer #5 (Remarks to the Author):

The manuscript reports the identification of candidate genes underlying the natural variation of photosynthesis efficiency in a diverse set of 344 *Arabidopsis thaliana* accessions. Photosynthetic efficiency was characterized through chlorophyll fluorescence imaging before and after a sudden increase in light irradiance. A pretty large number of candidate genes were reported by an initial GWAS analysis and the list was first narrowed down to 51 genes and then to 6 genes by making use of several criteria such as their expression and co-expression or their ontology and sequence variability between extreme phenotypes. The effect of 3 of these 6 genes was further confirmed using T-DNA lines which displayed a phenotype related to photosynthesis efficiency. In addition, one of these 3 genes, YS1, was validated by quantitative complementation. Finally, expression analyses and allele sequence comparisons further confirmed the effect of YS1 and pointed out sequence variation in its promoter that may account for the observed phenotypic variation.

The paper is interesting, it represents a large experimental effort and the results are pretty much convincing. I thus think they deserve to be published. Nevertheless I have some concerns with the way the manuscript is written because sometimes the story seems strange. Part of this may be due to some changes following the first round of review, especially with respect to the GWAS part. Indeed, I do believe that the multivariate methods are more appropriate to dynamically study a phenotype (which appeared to be on originality of the present paper with respect to previous work from the same team) than the univariate ones, but unfortunately the results from the multivariate analyses are not really well integrated to the manuscript as underlined by the fact that the initial list of 51 candidate genes was not modified following the multivariate analyses. One of the reason for this is likely because the gene YS1 was found only with the univariate approach and since the paper is focusing on this gene it may be difficult to account for the multivariate analyses results. Anyhow, I think this is a problem because it does not make the things easy to follow. Maybe this issue should be acknowledged and discussed in the manuscript.

⇒ *Response: The reviewer is right. We expected that a multi-trait GWAS would more clearly identify SNPs showing strong association with the phenotype. When this did not happen, but instead we got a much longer list of candidate genes, we decided to retain the initial list of prioritized candidate genes, also because many of these were identified in the multi-trait GWAS and we had initiated the screening of T-DNA knock-out mutants. The text has been modified to reflect our dilemma better. Although it would have been nicer if the multi-trait GWA analysis had identified SNPs close to YS1, we still think that in the end the GWA is nothing more than an exploratory tool to identify interesting regions (QTLs) and SNPs that can be further investigated by follow up experiments. Here the single time point GWA analyses brought us to the YS1, while the multi-trait analyses did not. The subsequent experiments prove the relevance of the YS1 gene, independent of the results of earlier GWA analyses.*

Also, YS1 was not found with a smaller window than 100 kb around the associated SNPs (method 2) because it is quite far away (100 kb) from the significant SNPs in QTL 23. I think that 100 kb on each side of the top SNP, which actually yields 200 kb QTL windows, is quite large given the expected LD decay of 10 kb in *Arabidopsis thaliana* (Kim et al., 2007). I admit that LD varies quite a lot in the genome especially because of natural selection but no information is given in the manuscript about LD extent around the QTLs and I clearly doubt that the LD decay would be the same within each of the 34 QTLs detected. Maybe LD decay could be estimated within each QTL to support the choice of 200 kb. If the authors want to underline their global approach as an original way to identify genes that matter for natural phenotypic variation, they should at least discuss these points.

⇒ *Response: We agree with the reviewer that LD varies a lot across the genome. We estimated local LD around the QTLs, following the approach of Mangin et al., 2012. Supplementary table S2 now shows the LD in terms of r^2 corrected for genetic relatedness, for each of the SNPs underlying a QTL. For a few QTLs LD appeared to extend to 50-100kb, while LD-decay was much*

faster for many other QTLs. We recognize that QTL-specific window sizes had been more appropriate here, but in the light of all the biological evidence presented for the function of YS1, we decided to stick to fixed 100 kb windows.

Finally, it is also unfortunate that the QTL 23 which harbor YS1 do not really show a dynamic response since it is detected both in low and high light (figure 2). If this is somehow consistent with the T-DNA results (figure 3), this is not really illustrative of the dynamic nature of the phenotype. Again, I do not have any fundamental problem with this but I think this should be mentioned and discussed in the text and maybe the goal of the paper should be modified.

⇒ *Response: That would indeed be unfortunately, but it is not the case! It is indeed not so easy to see from figure 2 (you may need to zoom in), but the locus which is seen at the third time point on the two days before the switch to high light is different from QTL23. Such is also indicated in the text and in supplementary table S3.*

Minor points:

- In the abstract it is stated "we show that it is feasible to dissect natural variation in photosynthesis efficiency down to the genomic DNA level by confirming that allelic sequence variation at the YELLOW SEEDLING1 (YS1) gene explains natural variation...". But in the end, how much of the natural variation is explained by YS1 alleles?

⇒ *Response: the YS1 locus explains at most 5.9 % of the natural variation, as indicated by Table S3 (using the formula $4\beta^2f(1-f)$, where the factor 4 is due to the fact that Arabidopsis is an inbreeder). For most time-points after the irradiance increase, this was around 5%; before the irradiance increase it was only 1.3%.*

- I would be interested by more quantitative genetics on the phenotypes at each time point. What are their broad and narrow (marker-based) heritability?

⇒ *Response: estimates of broad and narrow (marker-based) heritability are now provided in supplementary table S1 (B+C). Broad-sense heritability varied between 0.06-0.09 prior to stress and between 0.20-0.33 after stress. Marker-based estimates of narrow-sense heritability were close to zero prior to stress and varied between 0.30-0.52 after stress. For several time-points the latter values were larger than the broad-sense H2 estimates, but as explained in Kruijer et al. (2015) this can occur given the very large confidence intervals for narrow-sense heritability, that are typical for plant populations of this size.*

Is this phenotype structured within the population? Is there any correlation with the latitude of origin of the accessions? If so, are the alleles structured within the population under study? If this is the case I would expect some LD within the causal region which would support the choice of 200 kb for picking the candidate genes.

⇒ *Response: We examined this, but alas, we could not find any correlation between latitude and the phenotype.*

- Related to the potential structure of the phenotypes under study, I would be interested to see a QC of the GWAS through for instance QQ-plots or p-value distribution just to make sure that the potential confounding was well accounted for by the mixed-model.

⇒ *Response: QQ-plots are now provided in supplementary table S8. These clearly indicate that the mixed model adequately accounted for potential confounding.*

- There seems to be three time-replicates for GWAS and four time-replicates for F1, but in the material and methods this distinction is not made?

⇒ *Response: This is correct, it is added in the materials and methods section.*

Reviewers' Comments:

Reviewer #2 (Remarks to the Author):

The authors have considered my comments from the previous version and have address the concerns by adding in additional analysis. I have no further comments to add.

Reviewer #5 (Remarks to the Author):

The authors have adequately addressed my previous comments and they have modified the manuscript accordingly. The workflow for selecting candidate genes from the GWAS has been clarified and consequently this makes the manuscript easier to read.

Nevertheless, I still have very few minor points that would deserve further consideration:

1. The LD window used to select the candidate genes is of 100 kb on either side of the top SNPs resulting in 200 kb in total. Could you please add this precision (“on either side of the top SNPs”) in the text (195)?

2. In the supplementary table 3, I found some inconsistencies between the position of the top SNPs and the genomic location of the gene models within the corresponding LD window:

- QTL 9: the top SNP position is at 1.8 Mb on chromosome 5 while the genes are located between 17.6 and 17.8 Mb on the same chromosome;
- QTL18: the top SNP position is at 37.8 Mb on chromosome 2 while the genes are located between 7.3 and 7.5 Mb on the same chromosome;
- QTL20: the top SNP position is at 53.5 Mb on chromosome 2 while the genes are located between 3.3 and 3.5 Mb on the same chromosome.

Could you please check and fix these issues?

REVIEWERS' COMMENTS:

Reviewer #2 (Remarks to the Author):

The authors have considered my comments from the previous version and have address the concerns by adding in additional analysis. I have no further comments to add.

Reviewer #5 (Remarks to the Author):

The authors have adequately addressed my previous comments and they have modified the manuscript accordingly. The workflow for selecting candidate genes from the GWAS has been clarified and consequently this makes the manuscript easier to read.

Nevertheless, I still have very few minor points that would deserve further consideration:

1. The LD window used to select the candidate genes is of 100 kb on either side of the top SNPs resulting in 200 kb in total. Could you please add this precision ("on either side of the top SNPs") in the text (l 95)?

Response: this has been added

2. In the supplementary table 3, I found some inconsistencies between the position of the top SNPs and the genomic location of the gene models within the corresponding LD window:

- QTL 9: the top SNP position is at 1.8 Mb on chromosome 5 while the genes are located between 17.6 and 17.8 Mb on the same chromosome;

- QTL18: the top SNP position is at 37.8 Mb on chromosome 2 while the genes are located between 7.3 and 7.5 Mb on the same chromosome;

- QTL20: the top SNP position is at 53.5 Mb on chromosome 2 while the genes are located between 3.3 and 3.5 Mb on the same chromosome.

Could you please check and fix these issues?

Response: Thanks a lot for noting this! These issues have now also been fixed.